# The importance of active surveillance of carbapenem-resistant Enterobacterales (CRE) in colonization rates in critically ill patients

**Mabel Duarte Alves Gomides** [1,2]☯*, **Astrídia Marília de Souza Fontes** [3¤a‡], **Amanda Oliveira Soares Monteiro Silveira** [2¤b‡], **Daniel Chadud Matoso** [1¤a‡], **Anderson Luiz Ferreira** [2‡], **Geraldo Sadoyama** [2]☯

**1** Adult Intensive Care Unit, Uberlândia Clinical Hospital, Uberlândia, Brazil, **2** IBIOTEC (Institute of Biology and Technology), Federal University of Catalão, Catalão, Brazil, **3** Hospital Infection Control Sector, Uberlândia Clinical Hospital, Uberlândia, Brazil

☯ These authors contributed equally to this work.
¤a Current address: Jardim Umuarama, Uberlândia (Minas Gerais), Brazil
¤b Current address: Universitário, Catalão (Goiás), Brazil
‡ These authors also contributed equally to this work
* mabel@dermaclinicagoias.com.br

**Data Availability Statement:** All relevant data are within the paper.

**Funding:** The author(s) received no specific funding for this work.

## Abstract

### Objectivez

This study aimed to demonstrate the importance of active carbapenem-resistant Enterobacterales (CRE) surveillance and evaluate the prevalence of invasive infections, risk factors, and mortality risk in CRE-colonized patients.

### Methods

Retrospective cohort study analyzing 1,920 patients identified using an active CRE surveillance protocol, admitted to an adult intensive care unit in southeastern Brazil from January 2014 to December 2018.

### Results

There were 297 (15.47%) CRE colonized patients, with one colonized for every six control patients. CRE-colonized patients demonstrated an increased chance of infection (odds ratio [OR] 7.967, $p < 0.001$). Overall, 20.54% of the colonized patients presented invasive infection (81.96% due to Klebsiella pneumoniae). The colonization and infection ratio demonstrated the important role of the active CRE surveillance protocol. There were identified multiple risk factors for CRE colonization, including long-term mechanical ventilation (OR 1.624, $p = 0.019$) and previous exposure to aminopenicillins (OR 5.204, $p < 0.001$), carbapenems (OR 3.703, $p = 0.017$), cephalosporins (OR 12.036, $p < 0.001$), and fluoroquinolones (OR 5.238, $p = 0.012$). The mortality risk was significantly higher among colonized (OR 2.356, $p < 0.001$) and colonized-infected (OR 2.000, $p = 0.009$) patients and in those with *Enterobacter cloacae* colonization (OR 5.173, $p < 0,001$) and previous aminopenicillins exposure (OR 3.452, $p = 0.007$).

**Competing interests:** The authors have declared that no competing interests exist.

## Conclusions

Early detection of CRE colonization through screening testing proved to be an important tool to control CRE spread. However, observation over the years has shown no effective control of colonization and infection. The prevalence rates of CRE colonization and colonization-infection were high, as were the mortality rates. In conclusion, an active CRE surveillance protocol is essential, but its impact depends on the effective implementation of preventive measures and feedback between team members.

## Introduction

Colonization by carbapenem-resistant Enterobacterales (CRE) is an important cause of infection and one of the main sources of CRE dissemination in hospitals and in the community [1,2].

Dissemination of CRE has been occurring at an alarming pace worldwide due to the low response of these bacteria to available therapies and their difficult management with empirical antibiotics [3,4].

Consequently, there has been a high prevalence of CRE colonization and infection with serious threats to public health [5–10].

In asymptomatic carriers, the main CRE reservoir is the microbiota in the gastrointestinal tract, followed by the oropharynx, skin, and urine [11,12]. These colonization reservoirs are considered essential in the CRE spread inside and outside the hospital environment [11,13–16].

The genes determining Enterobacterales antimicrobial resistance have emerged in the past two decades [2,7,9] due to selective pressure caused by the wide use of broad-spectrum antibiotics, especially in intensive care units (ICUs) and long-stay hospitals [1,6,8,11,17,18].

Health-care units adopting specific antibiotic dispensing rules for each type of infection have individual microbial biosystems [13,17]. However, these treatments are not optimized and lead to increased resistance rates, including resistance to third generation cephalosporins (10–97%) [13,17]. This highlights the relevance of multidrug-resistance (MDR) bacteria in healthcare-associated infections (HAIs) [8,13,19–22].

Colonization by CRE can be acquired directly after antibiotic exposure or due to cross-transmission between patients and healthcare professionals [2,14]. The patients most susceptible to CRE colonization are those who are severely ill, have comorbidities, are subjected to invasive devices, and have previous antibiotic exposure, and long-term hospitalization [8,9,22,23].

Active CRE surveillance testing is an important strategy to control the spread of MDR bacteria, as it allows early implementation of protective measures through contact isolation, positively impacting public health [4,9,23–25]. The United States Centers for Disease Control and Prevention (CDC) considers the detection of CRE through rectal swabs the preferable method of CRE screening [25].

Patients with CRE colonization have a high probability of developing a subsequent infection that may be associated with bacteremia, leading to increased morbidity and mortality (30–75%) [7,8,10,17,24,26–28].

The occurrence of HAI caused by CRE is a serious matter since these infections frequently fail to respond to the delivered therapy, resulting in long-term hospital stay and increased hospital costs [10,13–15].

According to current CDC data, the mortality associated with antibiotic-resistant infections has decreased in the United States between 2013 and 2019 [29]. However, the number of registered infections caused by resistant bacteria remains high, with over 2.8 million cases/year and 35,000 deaths/year [29].

Based on these considerations, the aim of this study was to demonstrate the importance of active CRE surveillance in the ICU and verify the prevalence of CRE invasive infections in CRE-colonized and control patients. The study also aimed to determine the risk factors and risk of mortality related to strains, previous antibiotic exposure, and type of CRE colonization.

## Methods

### Study design

This was a retrospective cohort study including CRE-colonized patients, carried out in a 30 beds adult ICU of a public tertiary hospital in southeastern Brazil.

The study included patients older than 12 years, hospitalized in an adult ICU from January 2014 to December 2018. The patients were subjected to the active surveillance protocol for CRE detection and prevention established by the Hospital's Infection Control Commission (CCIH) in April 2011 (09/2011 Minutes). The resolution follows the guidelines from the Brazilian National Health Surveillance Agency (ANVISA) [30] and the criteria for antimicrobial resistance established by the World Health Organization (WHO) [31] and the CDC [25]. The active CRE surveillance protocol determines measures for detection with screening testing using rectal swabs collected upon the patient's admission and at weekly intervals during hospitalization, while the prevention measures are executed only for the cases confirmed of CRE colonization or infection. They include strict contact isolation in a properly identified with precaution contact signs private room, rapid and safe patient discharge, restricted use of probes and catheters, protection and hygiene of healthcare professionals, cleaning of the environment, and rational dispensing of antibiotics.

The research protocol was approved by the institution's Research Ethics Committee (protocol number 1.638.131).

### Data collection

A total of 2,126 active CRE surveillance protocol records were collected from 2012 to 2018. After review, patients' records who were discharged or died before the first rectal swab collection, as well as those who had incomplete or duplicated records and the period from 2012 to 2013 due to irregularities in the collections were excluded. After a final review of the protocol records, 1,920 patients subjected to the CRE surveillance protocol from January 2014 to December 2018 were included in the analysis. Demographic, clinical, and microbiological data from each individual record were collected, including the patients' identification record, age, sex, clinical diagnosis upon admission, disease severity score according to the Simplified Acute Physiology Score (SAPS) II, use of invasive devices, identification of Enterobacterales and other microorganisms, use of antimicrobials, results of rectal swabs for detection of CRE, and dates of admission and discharge or death.

The patients' records provided information regarding the length of ICU stay, use of long-term (over 10 days) mechanical ventilation, use of long-term (over 15 days) hemodialysis catheters, antibiotic exposure 30 days before the colonization and/or infection, and late death (30 days after admission). Some of the variables were stratified according to observed averages such as younger (13–54 years) and older (55–97 years) age groups, low (16–62) or high (63–217) disease severity scores (SAPS II), and short-term (1–21 days) or long-term (22–175 days) ICU stay.

The infected patients were clinically diagnosed with CRE infection and tested positive for CRE cultures, according to infection guidelines by the CDC's National Healthcare Safety Network (NHSN) [32]. The cultures of clinical and surveillance samples were subjected to identification and susceptibility tests using the VITEK2 automated system (BioMérieux, France) according to the Clinical and Laboratory Standard Institute (CLSI) guidelines [33]. The turn-around time mean of the CRE surveillance results was of five days. In case of positive clinical or surveillance culture for CRE, the microbiology laboratory informs immediately the medical or nursing staff of the ICU and the CCIH makes notifications in the medical electronic record and prescription.

## Statistical analysis

The clinical data are presented as frequencies. Continuous variables with normal distribution are presented as mean and standard deviation values and were compared using *Student's t* test. The normality of the distribution of quantitative variables was analyzed using the Shapiro-Wilk test. Pearson's chi-square test or Fisher's exact test was used to analyze the samples of colonized and non-colonized patients for infection and to determine the risk factors for colonization. The risk factors that were statistically relevant (p < 0.05) were submitted to a multivariate logistic regression test to determine the effect of all risk factors on the colonized and control patients, and to a multinomial logistic regression test to determine the influence of factors on CRE-colonized and colonized-infected patients in relation to controls. Multiple Cox regression, the Kaplan-Meier method, and the log-rank test were used to estimate survival by comparing curves between colonized, colonized-infected, and control groups. Multiple Cox regression was also used to calculate the proportional hazards of types CRE colonizers and antibiotic exposure previous to colonization. The adopted significance level (α) for all analyses was below 5% (p- value < 0.05). The strength of the association between the explanatory variables and the response was assessed by calculating the odds ratio (OR) with 95% confidence intervals (95% CIs). The analyses were carried out using the statistical software SPSS for Windows (IBM Corp., Armonk, NY, USA).

## Results

Among all 1,920 patients, the mean age was 52.42 ± 19.34 years (range 13–97 years), and there was a predominance of the male (65.31%) over the female (1.88:1) sex. Discharge (68.12%) was a more predominant outcome than death (2.13:1). The SAPS II score ranged from 16 to 131 (mean 62.19 ± 18.73). The mean length of hospital stay was 21.03 ±18.12 days (range 1–175 days).

A total of 3,154 rectal swabs were collected, of which 2,807 were negative for CRE in 1,623 patients (range 1–15 swabs/patient, mean 1.74 swabs/patient) and 344 were positive (10.91%) in 297 patients (range 1–3 positive swabs/patient, mean 1.16 swabs/patient). The most frequent strains identified were *Klebsiella pneumoniae* (83.16%), followed by *Enterobacter cloacae* (9.76%), *Escherichia coli* (4.38%), *Enterobacter aerogenes* (1.34%), *Serratia marcescens* (0.67%), *Enterobacter gergoviae* (0.34%), and *Serratia fonticola* (0.34%). Three patients had two rectal swabs each that isolated different CRE microorganisms; the first swab identified *Klebsiella pneumoniae* (in all three patients) and the second identified *Enterobacter cloacae* (in two patients) and *Serratia fonticola* (in one patients).

The patients were categorized into two groups: (1) colonized patients (n = 297, 15.47%), comprised by patients who tested positive in the CRE surveillance testing (including colonized-infected patients) and (2) control patients (n = 1,623, 84.53%), comprised by patients without CRE colonization (i.e., negative surveillance testing). In the colonized and control

patient groups, there was predominance of the male (66.32% and 65.12%, respectively) over the female sex (1.97:1 and 1.87:1, respectively) and more discharges (67.34% and 68.26%, respectively) in relation to deaths (2.06:1 and 2.15:1, respectively). The CRE-colonized group differed significantly from the control group in terms of SAPS II score (range 16–127, mean 65.54 ± 17.96 and range 16–131, mean 61.57 ± 18.80, respectively, p = 0.01), and length of hospital stay (interval 3–175 days, mean 33.40 ± 24.47 days and interval 1–145 days, mean 18.77 ± 15.68 days, respectively, p < 0.001) but not regarding age (range 14–97 years, mean 53.85 ± 19.37 years and range 13–93 years, mean 52.16 ± 19.33 years, respectively, p = 0.167).

Invasive CRE infections were detected in 61 (20.54%) of the 297 CRE-colonized patients and in 51 (3.14%) of the 1,623 control patients. Among the colonized-infected patients, the following CRE strains predominated in different collection sites: *K. pneumoniae* (81.96%) in urine, blood, tracheobronchial secretion, and tissue secretion; *E. cloacae* (16.39%) in blood, central venous catheter tip, and tissue secretion; and *E. coli* (4.92%) in urine and blood. Rates of other CRE strains were not significantly different in colonized-infected patients compared with controls, i.e., S. marcescens (1.64%) in urine, E. aerogenes (1.64%) in blood, and E. gergoviae (1.72%) in tracheobronchial secretion. Clinical infections caused by two strains in different cultures were identified in four patients colonized by *K. pneumoniae*; these included two patients with *K. pneumoniae* and *E. cloacae*, one patient with *K. pneumoniae* and *E. coli*, and one patient with *K. pneumoniae* and *E. aerogenes*. One patient colonized with *E. coli* was identified as having an invasive infection by *E. coli* and *E. cloacae*.

The infection to colonization ratio across the duration of the study were 1:3.5 in 2014, 1:2.4 in 2015, 1:3.6 in 2016, 1:8.3 in 2017, and 1:7.2 in 2018. Fig 1 shows the infected to colonized ratio for each year of the study.

Clinical invasive infections by the same colonizing CRE microorganisms occurred in 53 (17.84%) of the 297 colonized patients, of which 48 (90.56%) were K. pneumoniae. Analysis of CRE-colonized patients demonstrated an increased chance of infection with these strains (Table 1).

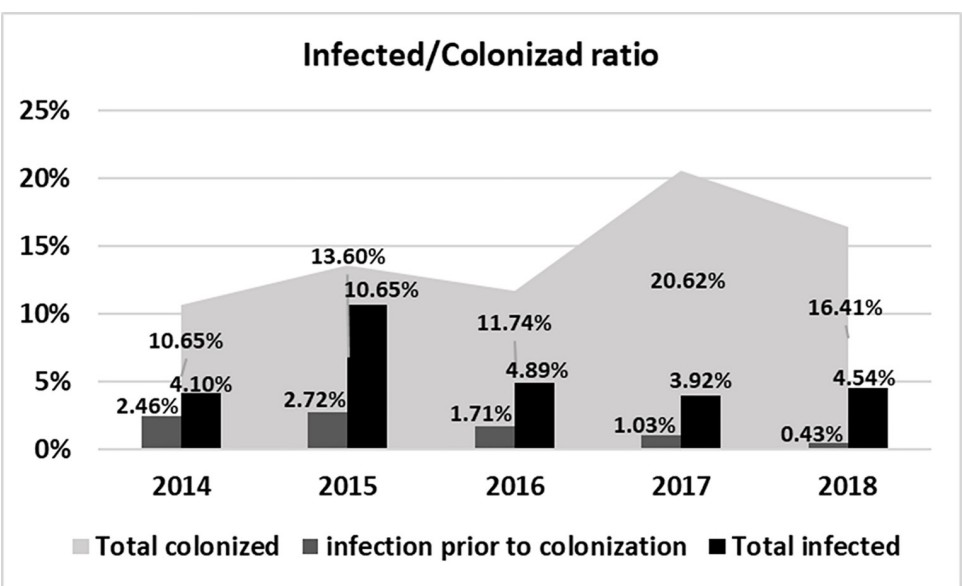

**Fig 1. Analysis of infection rates prior to colonization and of total colonized and total infected rates.**

**Table 1. Univariate analysis of the risk of development of invasive clinical CRE infection in the sample.**

| Infection | CRE colonized n (%) | Controls n (%) | OR (95% CI) | p |
|---|---|---|---|---|
| Yes | 61 (20.54) | 51 (3.14) | 7.967 (5.361–11.841) | <0.001 |
| No | 236 (79.46) | 1572 (96.86) | | |
| Total | 297 | 1623 | | |

CRE = Carbapenem-resistant Enterobacterales, n = number of patients, OR = odds ratio, 95% CI = confidence interval, p = test significance.

Risk factors for colonization were identified by assessing the effect of all independent variables that showed statistically significant differences (p < 0,05) in the comparative analysis of colonized and control patients (Table 2).

Following that, the association of these risk factors for colonization was analyzed by separately comparing the colonized and colonized-infected groups in relation to the control group. This final multivariable model showed also significant predictors for each group (Table 3).

**Table 2. Univariate analysis and multivariate logistic regression analysis of risk factors for CRE colonization in colonized versus non-colonized patients.**

| Variables | CRE colonized n (%) | Non-colonized n (%) | Bivariate | | Multivariate | |
|---|---|---|---|---|---|---|
| Total no. of patients | N = 297 (15.47) | N = 1623 (84.53) | OR (95% CI) | p | OR (95% CI) | p |
| Age range (55 to 97 Years) | 159 (53.54) | 804 (49.54) | 1.174 (0.916–1.504) | 0.208 | - | - |
| Sex (male) | 197 (66.33) | 1057 (65.13) | 0.948 (0.730–1.231) | 0.740 | - | - |
| SAPS (63 to 217) | 172 (57.91) | 759 (46.77) | 1.566 (1.220–2.012) | 0.001 | 0.733 (0.558–0.965) | 0.027 |
| ICU stays (22 to 175 days) | 175 (58.92) | 500 (30.81) | 3.222 (2.499–4.154) | <0.001 | 1.099 (0.762–1.585) | 0.614 |
| **Clinical diagnosis at ICU admission** | **n (%)** | **n (%)** | **OR (95% CI)** | **p** | **OR (95% CI)** | **p** |
| Septic shock | 26 (8.75) | 134 (8.26) | 1.066 (0.687–1.655) | 0.733 | - | - |
| Acute respiratory distress | 30 (10.10) | 123 (7.58) | 1.370 (0.900–2.085) | 0.161 | - | - |
| Neoplasm | 22 (7.41) | 99 (6.10) | 1.232 (0.762–1.989) | 0.366 | - | - |
| Postoperative | 26 (8.75) | 150 (9.24) | 0.942 (0.609–1.457) | 0.913 | - | - |
| Lower level of consciousness | 43 (14.48) | 270 (16.64) | 0.848 (0.599–1.202) | 0.393 | - | - |
| Infectious syndrome | 16 (5.39) | 51 (3.14) | 1.755 (0.987–3.121) | 0.059 | - | - |
| Inflammatory syndrome | 18 (6.06) | 95 (5.85) | 1.038 (0.617–1.745) | 0.893 | - | - |
| Metabolic syndrome | 20 (6.73) | 77 (4.74) | 1.450 (0.872–2.410) | 0.151 | - | - |
| Trauma | 96 (32.32) | 624 (38.45) | 0.765 (0.588–0.955) | 0.050 | - | - |
| **Invasive devices** | **n (%)** | **n (%)** | **OR (95% CI)** | **p** | **OR (95% CI)** | **p** |
| Central venous catheter | 290 (97.64) | 1474 (90.82) | 4.188 (1.942–9.030) | <0.001 | 1.972 (0.863–4.507) | 0.107 |
| Long-term hemodialysis catheter | 41 (13.80) | 91 (5.61) | 2,696 (1.822–3.989) | <0.001 | 1.225 (0.779–1.928) | 0.379 |
| Tracheostomy | 210 (70.71) | 638 (39.31) | 3.727 (2.848–4.877) | <0.001 | 0.617 (0.406–0.939) | 0.024 |
| Long-term MV | 218 (73.40) | 710 (43.75) | 3.548 (2.694–4.674) | <0.001 | 1.624 (1.085–2.430) | 0.019 |
| Long-term urethral catheter | 282 (94.95) | 1439 (88.66) | 2.404 (1.399–4.131) | 0.001 | 1.186 (0.646–2.177) | 0.583 |
| Enteral feeding tube | 268 (90.24) | 1266 (78.00) | 2.606 (1.746–3.891) | <0.001 | 1.315 (0.848–2.038) | 0.221 |
| Long-term gastric tube | 38 (12.79) | 115 (7.09) | 1.924 (1.303–2.841) | 0.002 | 1.023 (0.664–1.575) | 0.919 |
| **Previous antibiotic exposure** | **n (%)** | **n (%)** | **OR (95% CI)** | **p** | **OR (95% CI)** | **p** |
| Aminopenicillins | 14 (4.71) | 10 (0.62) | 7.980 (3.210–18.141) | <0.001 | 5.204 (2.244–12.066) | <0.001 |
| Carbapenems | 7 (2.36) | 7 (0.43) | 5.572 (1.940–16.005) | 0.003 | 3.703 (1.259–10.893) | 0.017 |
| Cephalosporins | 38 (12.79) | 14 (0.86) | 16.862 (9.011–31.555) | <0.001 | 12.036 (6.225–23.271) | <0.001 |
| Fluoroquinolones | 6 (2.02) | 4 (0.25) | 8.345 (2.341–29.755) | 0.002 | 5.238 (1.443–19.009) | 0.012 |

CRE = Carbapenem-resistant Enterobacterales, n = number of patients, OR = odds ratio, 95% CI = confidence interval; p = test significance, SAPS = Simplified Acute Physiology Score, MV = mechanical ventilation, ICU = intensive care unit.

**Table 3. Multinomial regression analysis of risk factors for patients with CRE colonization and colonization-infection compared with controls.**

| Variables | Colonized (n = 236) | | | Colonized-infected (n = 61) | | |
|---|---|---|---|---|---|---|
| | OR | 95% CI | p | OR | 95% CI | p |
| SAPS (63 to 217) | 1.433 | 1.070–1.921 | 0.016 | 1.127 | 0.638–1.990 | 0.681 |
| Central venous catheter | 2.061 | 0.859–4.946 | 0.105 | 1.467 | 0.175–12.296 | 0.724 |
| Long-term hemodialysis catheter | 0.980 | 0.588–1.632 | 0.937 | 2.490 | 1.206–5.141 | 0.014 |
| Tracheostomy | 1.472 | 0.935–2.316 | 0.095 | 2.703 | 0.976–7.489 | 0.056 |
| Long-term MV | 1.329 | 0.864–2.044 | 0.196 | 6.731 | 2.008–22.556 | 0.002 |
| Long-term urethral catheter | 1.067 | 0.576–1.975 | 0.836 | 2.756 | 0.448–31.467 | 0.222 |
| Enteral feeding tube | 1.348 | 0.847–2.148 | 0.208 | 1.098 | 0.364–3.313 | 0.868 |
| Long-term gastric tube | 0.873 | 0.536–1.421 | 0.585 | 1.621 | 0.797–3.296 | 0.182 |
| Aminopenicillins | 4.513 | 1.820–11.189 | 0.001 | 8.745 | 2.473–30.918 | 0.001 |
| Carbapenems | 2.565 | 0.728–9.034 | 0.143 | 9.223 | 2.125–40.037 | 0.003 |
| Cephalosporins | 8.363 | 4.085–17.122 | <0.001 | 35.021 | 14.224–86.222 | <0.001 |
| Fluoroquinolones | 3.230 | 0.708–14.742 | 0.130 | 15.114 | 3.012–75.842 | 0.001 |
| Long- term ICU stay | 1.365 | 0.914–2.038 | 0.128 | 0.447 | 0.209–0.955 | 0.038 |

CRE = Carbapenem-resistant Enterobacterales, n = number of patients, OR = odds ratio, 95% CI = confidence interval; p = test significance, SAPS = Simplified Acute Physiology Score, MV = mechanical ventilation, ICU = intensive care unit, n controls = 1623.

On multinomial logistic regression, there was a predominance of CRE strains in invasive devices of colonized-infected patients. The strains included K. pneumoniae in long-term hemodialysis catheters (OR 2.387, 95% CI 1.187–4.799, p = 0.015), tracheotomy (OR 3.262, 95% CI 1.093–9.733, p = 0.034), and long-term mechanical ventilation (OR 5.931, 95% CI 1.484–23.705, p = 0.012), as well as *E. cloacae* in long-term hemodialysis catheters (OR 4.624, 95% CI 1.274–16.782, p = 0.020).

Multiple *Cox* regression analysis was performed to evaluate the survival rate of patients with CRE the colonization by *K. pneumoniae* (OR 2.206, 95% CI 1.468–3.316, p < 0.001), *E. cloacae* (OR 5.173, 95% CI 2.372–11.281, p < 0.001), and other CRE (*E. aerogenes*, *E. coli*, *S. marcescens*, *E. gergoviae*, and *S. fonticola*) in relation to control patients. The other CRE bacteria were grouped since their individual analysis was not statistically significant (OR 1.398, 95% CI 0.640–3.057, p = 0.401) (Fig 2).

Multiple *Cox* regression analysis was also performed to evalute the mortality risk in CRE-colonized patients with previous antibiotic exposure (Fig 3). The mortality of patients who were colonized 30 days after use of antibiotics showed significant differences for the classes of aminopenicillins (OR 3.452, 95% CI 1.398–8.523, p = 0.007) and other grouped antibiotic classes such as carbapenems, cephalosporins, and fluoroquinolones (OR 1.829, 95% CI 1.078–3.102, p = 0.025), compared with the control group.

Survival probability analysis showed higher mortality in colonized-infected and colonized patients compared with controls (log-rank test p < 0.001) (Fig 4). Multiple *Cox* regression analysis of the variable "death" after 30 days of ICU stay among CRE-colonized patients showed a high mortality risk with significant differences in colonized (OR 2.356, 95% CI 1.547–3.587, p < 0.001) and colonized-infected (OR 2.000, 95% CI 1.187–3.368, p = 0.009) patients compared with controls.

## Discussion

Patients colonized by MDR bacteria are considered to be important reservoirs since they favor horizontal transmission of these microorganisms in the hospital environment [6,16].

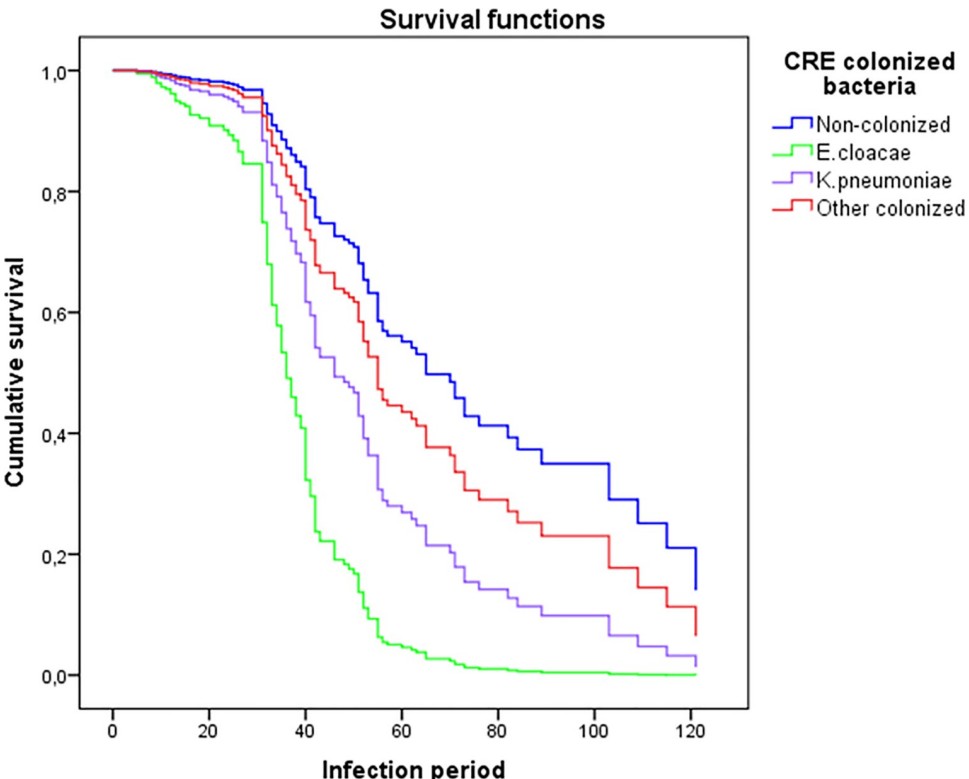

**Fig 2. Multiple Cox regression analysis of the estimated survival in patients with CRE-colonized bacteria compared with controls.**

Healthcare without safety standards for hand hygiene can amplify the risk of spreading resistant strains in hospital environments and threaten human health with an increased risk of infection-related morbidity and mortality [34].

In addition to the responsibility of healthcare teams, limited resources for CRE surveillance screening in underdeveloped countries is an additional concern for implementation measures for the detection of resistant bacteria since these delays the implementation measures for infection control and prevention [35,36]. There is still no consensus in the literature about the best method to detect CRE colonization, even though rectal swabs have been shown to be a sensitive method and with good correlation in the process of screening for active surveillance [15]. Based on that, rectal swabs are considered the main screening method for active surveillance [25].

In the literature, the positivity rate of CRE screening testing using rectal swab is about 10.1%, and the most frequent strains identified are *K. pneumoniae* (7.9–98.7%), followed by *E. cloacae* (22.0%), *E. coli* (20.0–82.1%), and other CRE (5.0%) [3,4,12,22,36–38]. In the present study, CRE screening testing carried out using rectal swab had a positivity rate of 10.91%, and the predominant strains were *K. pneumoniae* (84.17%), *E. cloacae* (9.76%), and *E. coli* (4.38%).

Multifaceted programs seeking control of intra-hospital CRE transmission have frequently targeted intervention measures by healthcare teams [39]. The literature has shown CRE colonization rates of 8.8% to 18.9% in patients in long-term units and 28% in transplant units [12,16,22,36]. The present study showed a high colonization rate (15.47%) in the sample, identifying at least one colonized patient for every six patients in the ICU.

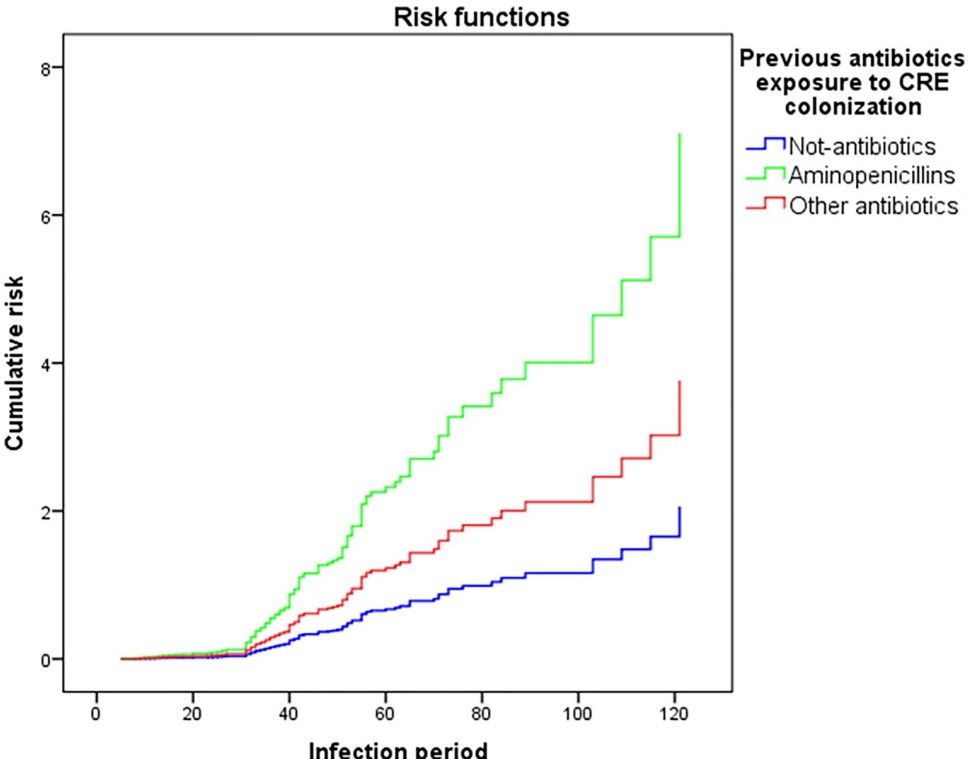

**Fig 3. Multiple Cox regression analysis of the risk estimation for previous antibiotics exposure in CRE-colonized patients compared with controls.**

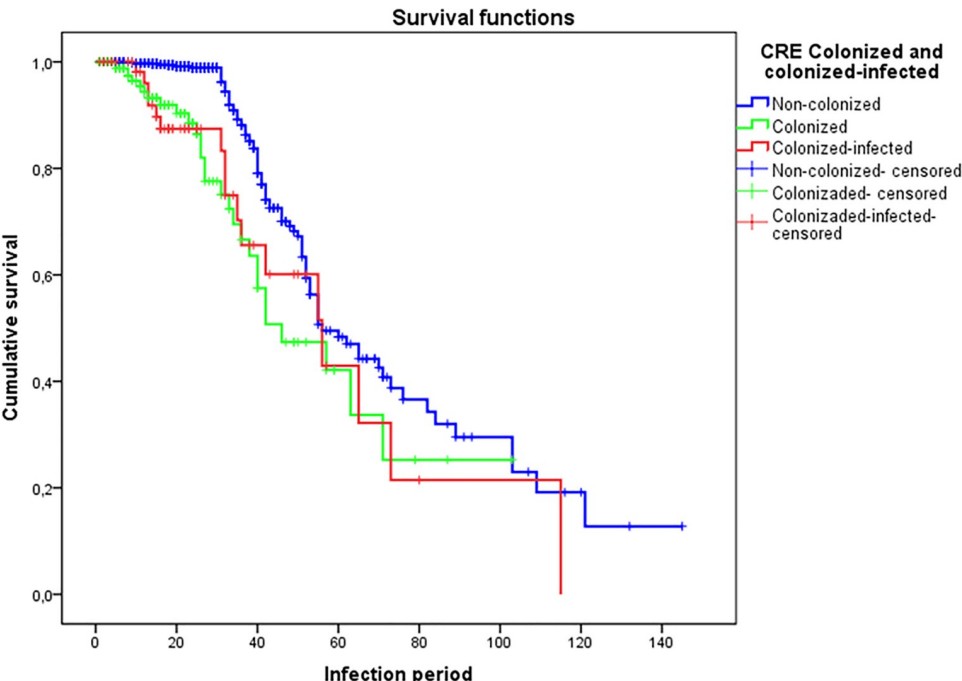

**Fig 4. Kaplan-Meier curves of estimated survival of colonized and colonized-infected patients compared with controls.**

The role of prevention of CRE infection and colonization goes beyond the identification of carriers through surveillance testing, as it also depends on continuous efforts to achieve multidisciplinary adherence to protective isolation, hand hygiene, environmental cleaning, and appropriate antibiotic dispensation [2,25].

In 2016, a study on safety culture analysis was conducted in the same ICU of the present study to evaluate the multidisciplinary team using the Safety Attitudes Questionnaire (SAQ) [40]. This safety culture study demonstrated reliable and significant results regarding safety perception but observed weak attitudes regarding management perception, working conditions, and communication failures [40]. The present study observed relevant data regarding colonization and infection rates and demonstrated the importance of active CRE surveillance for the implementation of early preventive measures. Even with these measures, the infection and colonization rates in the ICU were high during the study period.

Critically ill CRE-colonized patients are more prone to develop invasive infections with broad resistance to available antibiotics [28]. However, the disease's severity varies according to the pathogen's virulence and the host's potential defense and exposure to medical procedures [12,22,41]. According to the literature, CRE-colonized patients have an increased odd of developing infection (OR 2.06, p = 0.040), with rates ranging from 7.6 to 86.4% [3,7,10,12,26,27]. The infections developed by CRE-colonized patients have been predominantly by *K. pneumoniae*, followed by *E. coli*, *K. oxytoca* and *E. cloacae* [3,4,6,9,12,16]. In the present study, clinical CRE infection occurred in 20.54% of the CRE-colonized patients, corresponding to odds of infection of 7.967 (p < 0.001). *K. pneumoniae*, predominated in urinary, bloodstream, pulmonary, and subcutaneous tissue infections, while *E. cloacae* predominated in blood and pulmonary infections.

The factors that increase the risk of a critical patient becoming susceptible to CRE colonization are several and included tracheostomy (OR 4.8, p < 0.001), enteral feeding tube (OR 3.3, p = 0,001), long-term hospital stay (OR 3.8, p = 0.045), previous carbapenem exposure (OR 2.54, p < 0.05), and invasive procedures (OR 2.18, p < 0.05) [4,6,9,12,22,25]. The present study also revealed multiple risk factors for CRE colonization, including long-term mechanical ventilation (OR 1.624, p = 0.019) and previous exposure to aminopenicillins (OR 5.204, p < 0.001), carbapenems (OR 3.703, p = 0.017), cephalosporins (OR 12.036, p < 0.001), and fluoroquinolones (OR 5.238, p = 0.012).

Colonization by CRE is a pathogenic condition considered to be a strong determinant for the development of infection [3,6,22,26]. Some risk factors act as infection facilitators in colonized patients, including previous use of antibiotics such as fluoroquinolones (OR 3.04, p = 0.037), mechanical ventilation, non-surgical invasive medical procedures (OR 2.18, p < 0.05), endoscopy or colonoscopy (OR 3.7, p = 0.02), previous hospitalizations, and long-term ICU stay (OR 7.45, p = 0.023) [3,6,12,22,26,42]. In the present study, the risk factors for colonization and infection were similar but more prevalence in CRE colonized-infected patients compared with colonized and control patients. The risk factors in colonized-infected patients include long-term use of hemodialysis catheters (OR 2.490, p = 0.014), long-term mechanical ventilation (OR 6.731, p = 0.002), and previous exposure to aminopenicillins (OR 8.745, p = 0.001), carbapenems (OR 9.223, p = 0.003), cephalosporins (OR 35.021, p < 0.001), and fluoroquinolones (OR 15.114, p = 0.001).

The identification of modifiable risk factors and antibacterial resistance profile in hospital units brings positive results to protective measures, with a consequent reduction mortality [6,7,12]. Among the modifiable risk factors, stood out in the present study the predominance of infection by K. pneumoniae in invasive devices (81.96% in colonized-infected patients), including long-term mechanical ventilation, tracheotomy, and long-term hemodialysis catheters, and by *E. cloacae* (16.39% in colonized-infected patients) in long-term hemodialysis

catheters. The risk of mortality was significantly higher in patients colonized-infected by *E. cloacae* compared with other CRE.

Antibiotic dispensing programs determine the class of the antibiotic to be used, optimizing the antibiotic dose and limiting its duration of use [21,43]. These measures are helpful in HAI therapeutic responses and reducing the selection of drug-resistant microorganisms [44]. The importance of this appropriate control can be seen in the results of the present study, in which the mortality risk was significantly higher in patients with previous use of aminopenicillins and other antibiotic classes such as carbapenems, cephalosporins, fluoroquinolones, and polymyxins.

The most important adverse outcomes in CRE-colonized patients include infection and mortality, the latter with rates in the literature ranging from 27.5% to 41% [3,12,18]. The present study observed higher mortality in colonized-infected patients, followed by colonized ones.

## Conclusion

Early detection of CRE colonization through active CRE surveillance testing demonstrated to be important for implementing the necessary measures to contain the spread of these multiresistant microorganisms. However, observation over the years showed an absence of effective control of both CRE colonization and infection. The CRE-colonized patients had increased odds of infection, while colonized-infected patients had a higher risk of mortality. The occurrence of CRE colonization and colonization-infection presented known risk factors that can be modified through daily surveillance of signs of infection in catheters, daily awakening of the patients to disconnect mechanical ventilation, early removal of catheters and tubes, and implementation of antibiotic dispensing programs. In short, CRE active surveillance testing is essential, but its impact depends on effective actions in implementing preventive measures and feedback between team members.

## Acknowledgments

We thank professionals who participated in this study.

## Author Contributions

**Conceptualization:** Mabel Duarte Alves Gomides.

**Data curation:** Mabel Duarte Alves Gomides, Astrídia Marília de Souza Fontes.

**Formal analysis:** Mabel Duarte Alves Gomides.

**Methodology:** Mabel Duarte Alves Gomides.

**Resources:** Mabel Duarte Alves Gomides.

**Software:** Mabel Duarte Alves Gomides.

**Supervision:** Geraldo Sadoyama.

**Validation:** Mabel Duarte Alves Gomides, Anderson Luiz Ferreira.

**Visualization:** Amanda Oliveira Soares Monteiro Silveira, Daniel Chadud Matoso.

**Writing – original draft:** Mabel Duarte Alves Gomides.

**Writing – review & editing:** Mabel Duarte Alves Gomides.

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
