## [Decision Letter · Decision Letter 0]

6 Oct 2021

PONE-D-21-27215The importance of active surveillance of carbapenem-resistant Enterobacteriaceae (CRE) in colonization rates in critically ill patientsPLOS ONE

Dear Dr. Gomides,

Thank you for submitting your manuscript to PLOS ONE. After careful consideration, we feel that it has merit but does not fully meet PLOS ONE’s publication criteria as it currently stands. Therefore, we invite you to submit a revised version of the manuscript that addresses the points raised during the review process.

 Please provide clarifications on the points raised by the reviewers and correct the discrepancy in the dates mentioned in different sections of the manuscript.

We look forward to receiving your revised manuscript.

Kind regards,

Iddya Karunasagar

Academic Editor

PLOS ONE

Journal Requirements:

2. Please include in your Methods section the full name of the participating hospitals/institution.

Additional Editor Comments:

Two reviewers have commented on the manuscript and raised some very important questions and clarifications. Please address all comments point by point.

Reviewers' comments:

Reviewer's Responses to Questions

**Comments to the Author**

1. Is the manuscript technically sound, and do the data support the conclusions?

Reviewer #1: Yes

Reviewer #2: Yes

2. Has the statistical analysis been performed appropriately and rigorously? 

Reviewer #1: Yes

Reviewer #2: Yes

3. Have the authors made all data underlying the findings in their manuscript fully available?

Reviewer #1: Yes

Reviewer #2: Yes

4. Is the manuscript presented in an intelligible fashion and written in standard English?

Reviewer #1: Yes

Reviewer #2: Yes

5. Review Comments to the Author

Reviewer #1: The Aims and objectives to be clearly defined

Line 94 and 138- enterobacteria is it a standard terminology it must be Enterobacteriaceae

Invasive infections were found be more in colonized patients than the control group ,is it after application of all the preventive measures as per the hospital policy.

Line 228 and 231- to be written as association of risk factors and colonization not influence needs statistical consultation for using these terminologies

Reviewer #2: The manuscript highlights an important problem of gut colonization with CRE in a high risk population and I wish the best to the authors.

Please consider the following review comments :

1) Carbapenem resistant Enterobacteriaceae (CRE) should be replaced with "CR Enterobacterales" at all places including the title

2) Pg 5 : study design , line 115 : The method of processing the rectal swab must be elaborated for the benefit of the readers along with a reference

3) Pg 6 : data collection , line 130 mentions data collection from 2012 to 2018. Then Line 133: mention data included from 2014 to 2018. Please mention clearly the duration of the study

4) No clarity on the following issues :

• Wheter patients were kept in contact isolation till results arrived?

• What was the turn around time of the CRE surveillance results?

• What media were used for surveillance purposes?

6. PLOS authors have the option to publish the peer review history of their article (what does this mean?). If published, this will include your full peer review and any attached files.

Reviewer #1: No

Reviewer #2: **Yes: **Aruna Poojary

---

## [Author Response · Author response to Decision Letter 0]

19 Oct 2021

In response to the reviewers’ comments to the Author

Reviewer #1: The Aims and objectives to be clearly defined

Line 94 and 138- enterobacteria is it a standard terminology it must be Enterobacteriaceae

 Answer Line 94: I did not identify the enterobacteria word at line 94.

 Answer Line 138: enterobacteria was replaced by “Enterobacteriaceae”.

Invasive infections were found be more in colonized patients than the control group, is it after application of all the preventive measures as per the hospital policy.

 Answer: Yes, invasive infections in colonized patients occurred after application of all the preventive measures.

Line 228 and 231- to be written as association of risk factors and colonization not influence needs statistical consultation for using these terminologies

 Answer: It’s correct! The terminology was replaced after consult the statistical books.

Reviewer #2: The manuscript highlights an important problem of gut colonization with CRE in a high risk population and I wish the best to the authors.

 Answer: Thanks! 

Correction the following review comments:

1) Carbapenem resistant Enterobacteriaceae (CRE) should be replaced with "CR Enterobacterales" at all places including the title

Answer: "CR Enterobacterales" was replaced at all places including the title

2) Pg 5: study design, line 115: The method of processing the rectal swab must be elaborated for the benefit of the readers along with a reference

Answer pg 5: study design, line 115: The method of processing the rectal swab was written of the clear form on page 6: data collection, line 151, along with a reference.

3) Pg 6: data collection, line 130 mentions data collection from 2012 to 2018. Then Line 133: mention data included from 2014 to 2018. Please mention clearly the duration of the study

 Answer pg 6: data collection, line 130 and 133: The duration of the study was mentioned for best comprehension of the readers.

4) No clarity on the following issues:

• Whether patients were kept in contact isolation till results arrived?

 Answer: No, the patients weren’t kept in contact isolation till the results arrived. Therefore, the phrase of the pg. 5: study design, line 122 was reformulated for best comprehension the readers.

• What was the turnaround time of the CRE surveillance results?

 Answer: The turnaround time mean of the CRE surveillance results was of five days. These data were described on pg. 7: data collection, line 154, for best comprehension of the readers.

• What media were used for surveillance purposes?

 Answer: The media used for surveillance purposes were information immediate of the microbiology laboratory and notifications in the electronic record and prescription medical by Hospital Infection Control Sector. These data were described on pg. 7: data collection, line 155, for best comprehension of the readers.

---

## [Decision Letter · Decision Letter 1]

22 Nov 2021

PONE-D-21-27215R1The importance of active surveillance of carbapenem-resistant Enterobacterales (CRE) in colonization rates in critically ill patientsPLOS ONE

Dear Dr. Gomides,

Thank you for submitting your manuscript to PLOS ONE. After careful consideration, we feel that it has merit but does not fully meet PLOS ONE’s publication criteria as it currently stands. Therefore, we invite you to submit a revised version of the manuscript that addresses the points raised during the review process.

Two minor comments from Reviewer 2 to be addressed. 

We look forward to receiving your revised manuscript.

Kind regards,

Iddya Karunasagar

Academic Editor

PLOS ONE

Journal Requirements:

Additional Editor Comments:

Please see two minor comments from Reviewer 2.

Reviewers' comments:

Reviewer's Responses to Questions

**Comments to the Author**

1. If the authors have adequately addressed your comments raised in a previous round of review and you feel that this manuscript is now acceptable for publication, you may indicate that here to bypass the “Comments to the Author” section, enter your conflict of interest statement in the “Confidential to Editor” section, and submit your "Accept" recommendation.

Reviewer #1: All comments have been addressed

Reviewer #2: All comments have been addressed

2. Is the manuscript technically sound, and do the data support the conclusions?

Reviewer #1: Yes

Reviewer #2: Yes

3. Has the statistical analysis been performed appropriately and rigorously? 

Reviewer #1: Yes

Reviewer #2: Yes

4. Have the authors made all data underlying the findings in their manuscript fully available?

Reviewer #1: Yes

Reviewer #2: Yes

5. Is the manuscript presented in an intelligible fashion and written in standard English?

Reviewer #1: Yes

Reviewer #2: Yes

6. Review Comments to the Author

Reviewer #1: All the comments are addressed and incorporated in the manuscript. The study is an important research work adds to the medical literature the relevance of looking for multi drug resistant bacterial colonizers and implement preventive measures to control their spread in the critical areas of the health settings

Reviewer #2: Thank you for clarifying the comments. Pls find below the comments from the fresh review

1) Pg 66 , line 73 : Capital E for Enterobacterales

2) Pg 69 line 141: pls change to Enterobacterales

7. PLOS authors have the option to publish the peer review history of their article (what does this mean?). If published, this will include your full peer review and any attached files.

Reviewer #1: No

Reviewer #2: **Yes: **Aruna Poojary

---

## [Author Response · Author response to Decision Letter 1]

24 Dec 2021

Rebuttal Letter

In response to the reviewers’ comments to the Author

Items 1 to 5 - compliance between reviewers and article

6. Review Comments to the Author

Reviewer #1: All the comments are addressed and incorporated in the manuscript. The study is an important research work adds to the medical literature the relevance of looking for multi drug resistant bacterial colonizers and implement preventive measures to control their spread in the critical areas of the health settings

Answer: Thanks your comments!

Reviewer #2: Thank you for clarifying the comments. Pls find below the comments from the fresh review

1) Pg 66, line 73: Capital E for Enterobacterales

Answer Line 73: enterobacterales was replaced by “Enterobacterales”.

2) Pg 69 line 141: pls change to Enterobacterales

Answer Line 138: Enterobacteriaceae was replaced by “Enterobacterales”.

7. PLOS authors have the option to publish the peer review history of their article (what does this mean?). If published, this will include your full peer review and any attached files.

Do you want your identity to be public for this peer review? For information about this choice, including consent withdrawal, please see our Privacy Policy.

Reviewer #1: No

Answer: Thanks your review in this article!

Reviewer #2: Yes: Aruna Poojary

Answer: Thanks your review in this article!

---

## [Editor Report · Decision Letter 2]

29 Dec 2021

The importance of active surveillance of carbapenem-resistant Enterobacterales (CRE) in colonization rates in critically ill patients

PONE-D-21-27215R2

Dear Dr. Gomides,

We’re pleased to inform you that your manuscript has been judged scientifically suitable for publication and will be formally accepted for publication once it meets all outstanding technical requirements.

Kind regards,

Iddya Karunasagar

Academic Editor

PLOS ONE

Additional Editor Comments (optional):

All reviewer comments have been addressed.
---

## [Editor Report · Acceptance letter]

7 Jan 2022

PONE-D-21-27215R2 

The importance of active surveillance of carbapenem-resistant Enterobacterales (CRE) in colonization rates in critically ill patients 

Dear Dr. Gomides:

I'm pleased to inform you that your manuscript has been deemed suitable for publication in PLOS ONE. Congratulations! Your manuscript is now with our production department. 

Kind regards, 

on behalf of

Dr. Iddya Karunasagar 

Academic Editor

PLOS ONE